

# Modelling non-stationary flood frequency in England and Wales using physical covariates

Duncan S. Faulkner[1], Sean Longfield[2,3], Sarah Warren[1] and Jonathan A. Tawn[4]

[1]JBA Consulting, Skipton, BD23 3FD, UK
[2]Environment Agency, Leeds, LS11 9AT, UK
[3]Department of Geography, University of Lincoln, Lincoln, LN6 7TS, UK
[4]Department of Mathematics and Statistics, Lancaster University, Lancaster, LA1 4YF, UK

Correspondence to: Duncan Faulkner (duncan.faulkner@jbaconsulting.com)

**Abstract**

Non-stationary methods of flood frequency analysis are widespread in research but rarely implemented by practitioners who manage flood risk. One reason for this may be that research papers on non-stationary statistical models tend to focus on model fitting rather than extracting the sort of results needed by designers and decision makers. It can be difficult to extract useful results from non-stationary models that include stochastic covariates for which the value in any future year is unknown. Examples of such covariates include rainfall, temperature or indices of fluctuations of atmospheric pressure.

We explore the motivation for including such covariates, whether on their own or in addition to a covariate based on time. We set out a method for expressing the results of non-stationary models, and their uncertainty, as an integrated flow estimate, which removes the dependence on a particular value of the covariates. This can be defined either for a particular year or over a longer period of time. The methods are illustrated by application to a set of 375 river gauges across England and Wales. We find annual rainfall to be a useful covariate at many gauges, sometimes in conjunction with a time-based covariate.

For estimating flood frequency in future conditions, we advocate exploring hybrid approaches that combine the best attributes of non-stationary statistical models and simulation models that can represent the impacts of changes in climate and river catchments.

## 1 Introduction

As the impacts of environmental change become increasingly evident in time series, non-stationary methods of frequency

analysis have become widespread in research. Examples include analysis of rainfall frequency, extreme sea levels and fluvial flood frequency, the primary focus of this paper.

Application of non-stationary methods by practitioners involved in engineering and environmental management is much more limited. We surveyed practitioners and flood management authorities in seven countries, asking whether their currently recommended methods of flood and/or rainfall frequency estimation accounted for non-stationarity over the period of

measurements (Luxford and Faulkner, 2020). Apart from methods that account for sea level rise, non-stationary methods of





flood frequency estimation are not mandated by flood management authorities or generally used by practitioners in the USA, Canada, Australia, the Netherlands, Germany or (until 2020) the UK. The only example that we found of them being used by practitioners before 2020 was in Switzerland, where the Federal Office for the Environment has fitted a range of non-stationary models to peak flow data from many catchments (BAFU, 2017).

There are many possible reasons why research is not implemented in practice. Here we focus on only one: the difficulty in extracting useful results from some non-stationary statistical models of flood frequency. Research papers tend to focus on fitting non-stationary statistical models rather than extracting the sort of results needed by designers and decision makers.

There is a particular difficulty in extracting results from models that use some types of physical variables as covariates – in other words, the parameters of the model are linked to stochastic variables such as rainfall or climatic indices. This is well

illustrated in a paper by Hesarkazzazi *et al*. (2021) who present results from non-stationary models of flood frequency in north-west England that show jumps from year to year, conditional on the accumulated rainfall during the year. Results such as these cannot be directly used in flood risk management, because every year the rainfall will be different. The same authors identified a need for further research into defining a frequency distribution for the covariates when introducing an extra stochastic component into a model. Similarly, Šraj *et al*. (2016) present a model with annual precipitation as a covariate but do not attempt

to account for the distribution of that covariate, merely suggesting that the model allows estimation of design flow for a specific value of annual precipitation that is likely to occur over the lifetime of a structure. In a third example, Chen, Papadikis *et al*. (2021) present quantiles for non-stationary models plotted against various covariates, without commenting on how such results could be interpreted to make them applicable in design and assessment of flood management measures.

There is also a need to quantify the uncertainty associated with the results of non-stationary models, which can be expected to

be larger than the uncertainty of stationary estimates of flood frequency (Serinaldi and Kilsby, 2015).

In this paper we first outline the motivation for including physical covariates in non-stationary models of flood frequency. We then set out methods of applying such models that can provide the type of information needed by practitioners. We illustrate these new approaches with an application in England and Wales. The research described here formed part of a project funded by the Environment Agency in England that produced methods, tools and guidance to equip practitioners to carry out non-

stationary analysis of fluvial flood frequency (Faulkner *et al.,* 2020).

**Non-stationary flood frequency models including physical covariates**

The method described here is based on fitting frequency distributions to annual maximum river flows. Two distributions have been applied, each with three parameters: the Generalised Extreme Value (GEV) and the Generalised Logistic (GLO). The connection between these distributions is explained by Eastoe and Tawn (2010). The two distributions can give increasingly

different estimates for low-probability floods (Hesarkazzazi *et al*., 2021).

The distributions are fitted using maximum likelihood estimation (MLE).

The GEV distribution function is of the form





$$F(x) = exp\left\{-\left[1 + \xi\left(\frac{x-\mu}{\sigma}\right)\right]_{+}^{-1/\xi}\right\} \tag{1}$$

and the GLO is of the form:

$$F(x) = \frac{1}{1+\left[1+\xi\left(\frac{x-\mu}{\sigma}\right)\right]_{+}^{-1/\xi}} \tag{2}$$

where for both distributions $\mu$ is the location parameter, $\sigma$ is the scale parameter, $\xi$ is the shape parameter, $y_{+}= \max\{y, 0\}$, $\xi \neq 0$ and $\sigma > 0$.

For the sake of simplicity, it was assumed that $\mu$ and log $\sigma$ vary linearly with covariates, with $\xi$ modelled as a constant. For a vector of covariates $x$:

$$\mu(x) = \mu_0 + \mu_1 x_1 + \mu_2 x_2 + \cdots \tag{3}$$

$$\sigma(x) = \exp(\phi_0 + \phi_1 x_1 + \phi_2 x_2 + \cdots). \tag{4}$$

Thus for a non-stationary fit there are two or more elements of the location parameter to estimate, a constant component $\mu_0$ and $\mu_1$, $\mu_2$, etc. which represent the influence of the covariates on the parameter, and the same for the scale parameter.

The most common approach to non-stationary flood frequency estimation is to model the non-stationarity only as a function of time (Hesarkazzazi *et al.*, 2021). Recent examples include Prosdocimi and Kjeldsen (2021) and Griffin *et al.* (2019). However, many authors include physical quantities as covariates such as annual rainfall (Sraj, 2016; Yan *et al.*, 2017; Hesarkazzazi *et al.*, 2021; Chen, Papadikis *et al.*, 2021), extreme rainfall (Prosdocimi *et al.*, 2014), temperature (Hesarkazzazi *et al.*, 2021; Wasko, 2021), urban extent (Prosdocimi *et al.*, 2015), population (Yan *et al.*, 2017) or climatic indices such as the North Atlantic Oscillation (NAO) (Steirou *et al.*, 2019), East Atlantic pattern (EA) (Francois *et al.*, 2019), El Niño Southern Oscillation (El Adlouni *et al.*, 2007) or Interdecadal Pacific Oscillation (Franks *et al.*, 2015).

One of two reasons is typically given for modelling non-stationarity of floods using physically based covariates:

1. Physical covariates help remove some of the year-to-year variability in AMAX flows, enabling better identification of time-based trends and better fit of the distribution (Prosdocimi *et al.*, 2014; Hesarkazzazi *et al.*, 2021).

2. They provide a more physically meaningful model of non-stationarity, since time of itself has no physical influence on flooding (Chen, Papadikis *et al.*, 2021). As a covariate, time is merely a substitute for some other physical quantity that is influencing floods. Some physical covariates may open up the prospect of predicting the future evolution of the flood frequency curve (Sraj *et al.*, 2016).

Reason (1) leads to models that include both time and physical quantities as covariates. One important consideration is the effect of collinearity and confounding variables on results. The greater the dependence between covariates then the harder it is to interpret the regression coefficients and the more numerical convergence issues that will arise. To reduce these problems, it is desirable if the covariates are orthogonal (i.e. uncorrelated). When time is one of the covariates, this can be achieved by detrending the physical covariates before inclusion in the statistical model to ensure that they are not correlated with time. The time covariate will then represent the presence of any temporal trend in the flood peak series.





Reason (2) tends to lead to a rather different approach in which the physical variables replace time as a covariate. Within this
approach, to model a flood series that has a trend over time then it would be preferable to include at least one physical covariate
in the model which exhibits a time trend. If this approach is to be used to understand future changes in flood risk, there is a
need to model the trend in that covariate. The hope here is that a covariate can be found for which the trend is easier to model
than that in the flood series, perhaps because it has less variability and more predictability into the future. An example might
be the extent of urbanisation in a catchment, which can be typically expected to show a monotonic increase over time.
Additionally, urbanisation can be reasonably predicted into the future under a range of scenarios.

A risk associated with this second approach is the confusion of correlation for causation. In principle it would be possible to
include any covariate with a trend, whether or not it had any physical connection with the processes that cause floods. This
can lead to a false sense of confidence about our ability to estimate the future evolution of the flood frequency curve: we might
end up with a covariate for which we can confidently predict future values, but which is no more useful than the date as a way
of explaining observed trends in flood magnitudes. We return to this theme in the Discussion.

This potential for misinterpretation has led some authors to criticise the move to non-stationary frequency modelling, or to
express caution. The arguments are summarised in Faulkner *et al*. (2019).

## 2 Extracting results from non-stationary flood frequency models

Practitioners tend to refer to the quantiles, or return levels, of a flood frequency distribution as flood frequency estimates or
design floods. These estimates are associated with a defined probability of exceedance over a defined period of time. It is
common to refer to annual exceedance probability, the inverse of which is the return period on the annual maximum scale.
These concepts become unwieldy in a non-stationary setting, and various alternative definitions and measures of probability
have been proposed. For planning of investments in flood protection or decisions about permitting development of land, an
annual probability of flooding is usually less relevant than the probability of flooding during a longer design horizon, such as
the expected design life of a flood defence scheme. We use the concept of the **encounter probability**, which is the probability
of an event occurring over a defined number of years. It is equivalent to the design life level, proposed by Rootzén and Katz
(2013), which quantifies the risk of a flood exceeding a threshold value during a given period such as the design life of a
structure.

Flood frequency estimates from a non-stationary distribution will change over time, if time is included as a covariate, and will
also depend on any physical covariates. For instance, if the covariate is annual rainfall, then the flow with a 1% annual
exceedance probability (AEP) given 1200mm of rainfall is the expected flow under the (clearly hypothetical) conditions that
the annual rainfall is always 1200mm. We refer to this quantity as the **conditional flow estimate**.

The conditional flow estimate may be useful when examining the probability of past floods, but it is less informative when
thinking about design and planning for present-day or future floods. We introduce the term **integrated flow estimate**, which
removes the dependence on a particular value of the covariates. It is defined as the return level corresponding to the encounter
probability averaged over covariates in a period of interest. The calculation method is set out below. The concept was





introduced by Eastoe and Tawn (2009), who refer to it as the marginal return level. The need for further methodological development of this concept was identified by Faulkner *et al.* (2019).

This section sets out methods for calculating both conditional and integrated flow estimates, in each case for a non-exceedance probability value of $p$.

Let $F(y|x; \theta)$ be the conditional distribution function of annual maximum flow, where $x$ is a vector of covariate values for the year in which the annual maximum is considered, and $\theta$ is the vector of parameters of the distribution, including the coefficients that relate the distribution parameters to the covariates.

Let $y_p(x)$ be the conditional flow estimate for probability p (the conditional pth quantile), i.e. conditional on a particular set of values $x$ for the covariates.

Let $F(y; \theta)$ be the distribution function for annual maximum flow in the period of interest (e.g., current or future time window typically exceeding a year in duration – such as over a design lifetime), where $\theta$ is the vector of parameters of the distribution. This distribution function does not depend on the covariate values of the particular year in which the annual maximum occurs but incorporates the distribution of the covariates over the period of interest.

Let $y_p$ be the integrated flow estimate for probability $p$ (the marginal $p$th quantile).

Let $f(x)$ be the joint density function of the covariates for the time period of interest, so this distribution could change depending on whether the period of interest is the current or a future time window.

Let $\Phi$ be the domain of the covariates.

$F(y; \theta)$ can be obtained from the conditional distribution function $F(y|x; \theta)$ by integrating out the covariates:

$$F(y; \theta) = \int_\Phi F(y|x; \theta) f(x) \, dx. \tag{5}$$

The integrated flow estimate for the period of interest is then $y_p$, which is such that $F(y_p; \theta) = p$.

The integrated flow estimate for a year in the period of interest is obtained by inverting the distribution function: $y_p = F^{-1}(p; \theta)$. This needs to be solved numerically.

In practice, the parameters $\theta$ of this distribution are not known, nor is the true density $f(x)$ of the covariates, so to get an integrated flow estimate, they need to be replaced with estimates in Equation 5. Here, $\theta$ is estimated by $\hat{\theta}$, the maximum likelihood estimate of the parameters, and the joint density of the covariates $f(x)$ needs to be replaced by some estimate $\hat{f}(x)$, for example, the empirical density of the data or a kernel density estimate, which smooths the empirical estimate (Silverman, 1998). The data used to construct this kernel density estimate depends on the period of interest.

The integrated flow estimate for probability $p$, $\hat{y}_p$, is found from $\hat{y}_p = \hat{F}^{-1}(p; \hat{\theta})$.

Similarly, the conditional flow estimate for probability $p$ is obtained from $\hat{y}_p(x) = \hat{F}^{-1}(p|x; \hat{\theta})$.

The integrated flow estimate should be understood as applying over a period rather than instantaneously. This is a useful concept for planning investment decisions in flood risk management, which need to consider the probability of floods occurring over the period of the planning horizon.





The integrated estimate is distinct from the stationary estimate of a design flood, even though the two might appear superficially

similar because both will plot as horizontal lines on a time series graph. The integrated flow estimate could also be calculated separately for different portions of the record. It is also possible to calculate the integrated flow estimate by integrating over a sample of covariates that spans a period different from that covered by the river flow data. For instance, the record of the covariates might be longer, enabling a more confident estimate of the distribution of the covariates.

In theory, it could also be calculated by averaging over a distribution of covariate values intended to represent future conditions.

This is not necessarily suitable for practical application because it is only valid if the physical covariates provide a complete causal description of the non-stationarity in peak flows. If this approach were to be applied, it would be possible to derive a flow estimate corresponding to the expected annual exceedance probability over the design period. This is equivalent to the 'average design life level' presented in Yan *et al*. (2017). The output from the analysis is akin to a probability of a particular flood flow being exceeded during the lifetime of a scheme or a development.

If the covariates include both time and physical variables, it is possible to calculate an integrated flow estimate by averaging the probabilities corresponding to the observed physical covariate values, but setting the time covariate to a single value, such as the final year of record. This gives what we term a **single-year integrated flow estimate**. If the river flow record runs up to the present day, this estimate is representative of the flow expected to be exceeded with a given probability under current conditions, without being conditional on any particular value of a covariate. The single-year integrated flow estimate can be

more easily compared with alternative estimates such as those from a model that uses only time as a covariate.

We present a method for calculating confidence limits of the estimates in Appendix A.

## 3 Application in England and Wales

The method described above was applied to annual maximum flow series at a set of 375 river gauging stations in England and Wales. The dataset was screened to eliminate gauges where apparent non-stationarity might arise from changes in the flow

measurement method or the local hydraulics. Details of the screening and of the final dataset are given in Faulkner *et al.* (2020). The range of record lengths was from 27 to 134 years, with a median of 48 years.

In light of findings from the literature, summarised in Faulkner *et al.* (2020), the following seven variables were selected as trial covariates:

- Catchment-average rainfall, calculated over the water year (October to September), the autumn and the winter

185         seasons. Rainfall accumulations were calculated from the CEH-GEAR dataset which provides daily rainfall on a 1km grid across the UK from 1890 (Tanguy *et al*., 2016).

- North Atlantic Oscillation (NAO) index, averaged over the winter, summer and autumn.
- East Atlantic pattern (EA) index, averaged over the winter.

Although some studies have used indices of extreme rainfall as covariates (e.g. Chen, Papadikis *et al*., 2021), we excluded

them to avoid shifting the problem into estimating the frequency distribution of an extreme value elsewhere in the hydrological





cycle. We also considered only covariates that are expected to be significant across many catchments in preference to those that represent locally specific effects such as urbanisation or changes in forest cover.

NAO and EA indices were obtained from NOAA (2019). The rainfall data were centred and scaled, subtracting the mean from each observation and dividing the result by the standard deviation. This transformation is expected to reduce computational problems in which the likelihood optimisation algorithm converges to local maxima; each of the covariates is essentially on the same scale, so the regression parameters are more comparable over covariates for numerical purposes. The other covariates were already standardised in a similar way.

The physical covariates were included in candidate non-stationary models in accordance with the following: Up to 2 covariates per model, with a maximum of one being a physical covariate, the other being water year, with the physical covariate being detrended for reasons given earlier. Every possible combination was included in a candidate model, resulting in 88 candidate models for each gauging station, which were then compared. The candidate models therefore included:

       (1) a stationary version,

       (2) versions with just time as a covariate,

       (3) versions with just one detrended physical covariate, and

       (4) versions with both time and a detrended physical variable.

Covariates were considered for modelling either or both of the location and scale parameters. The model fitting was repeated for (i) the GEV and (ii) the GLO distribution. For both cases the results were output from the best-fitting model, judged by the lowest Bayesian Information Criterion (BIC).

Detrending was carried out even when fitting model (3) because for the model comparison to be valid it is necessary to be fitting all models to equivalent covariates. However, it would seem odd if year-to-year variations were captured by a detrended physical covariate without the longer-term changes in that covariate (before detrending) also being important. To better understand the interaction between time and physical covariates, an additional set of model versions was fitted at sites where model (4) fitted best:

       (5) versions with just one physical covariate, as measured (without detrending).

If version (5) performs better than version (4) then this may indicate that the same physical covariate is capturing both inter-annual variations and a longer-term trend.

Further details of the covariate screening and model selection procedure are given by Faulkner *et al.* (2020). The calculations used the R package nonstat (Warren and Longfield, 2020). Results are presented for the GEV here.

## 4 Results

Faulkner *et al.* (2020) provide a detailed account of most of the results. Here we focus on results from models using physical covariates. At nearly all gauges (97%), models including physical covariates provided the preferred fit, according to the BIC. The majority of these were type (3): models without time as a covariate (Table 1). The findings for these gauges, 61% of the



total, indicate that flood magnitude is statistically associated with the value of the (detrended) physical covariate, without any temporal non-stationarity. At 36% of gauges, the preferred model was type (4), combining a physical quantity with time as covariates.

**Table 1: Proportions of model types preferred across the dataset (GEV distribution)**

| Model type | (1) Stationary: No covariates | (2) Non-stationary: time the only covariate | (3) Temporally stationary with detrended physical covariates | (4) Non-stationary: time and detrended physical covariates |
|---|---|---|---|---|
| % of gauges | 2% | 1% | 61% | 36% |

At only 1% of gauges is a model with only water year as a covariate preferred. This finding indicates that physical covariates are adding useful information in nearly all cases, even though their linear trends have been removed. This is an important finding which hints at a causal relationship between the physical covariates and flood flows.

The increase in model complexity is outweighed by the increase in goodness of fit provided by the (detrended) physical covariates. Figure 1 shows that the de-trended physical covariates give a bigger improvement in fit than using time only, with a median improvement in the BIC value of almost 15 for models (3) and (4) compared to model (1). The equivalent when comparing model (2) with model (1) is less than 5. This indicates that the greatest source of year-to-year variation is caused by physical covariates (ignoring their linear trends), not simply time, again hinting at a causal relationship. It also shows that regressing against covariates which aim to pick up linear trends ignores more statistically important additional sources of inter-annual variation.





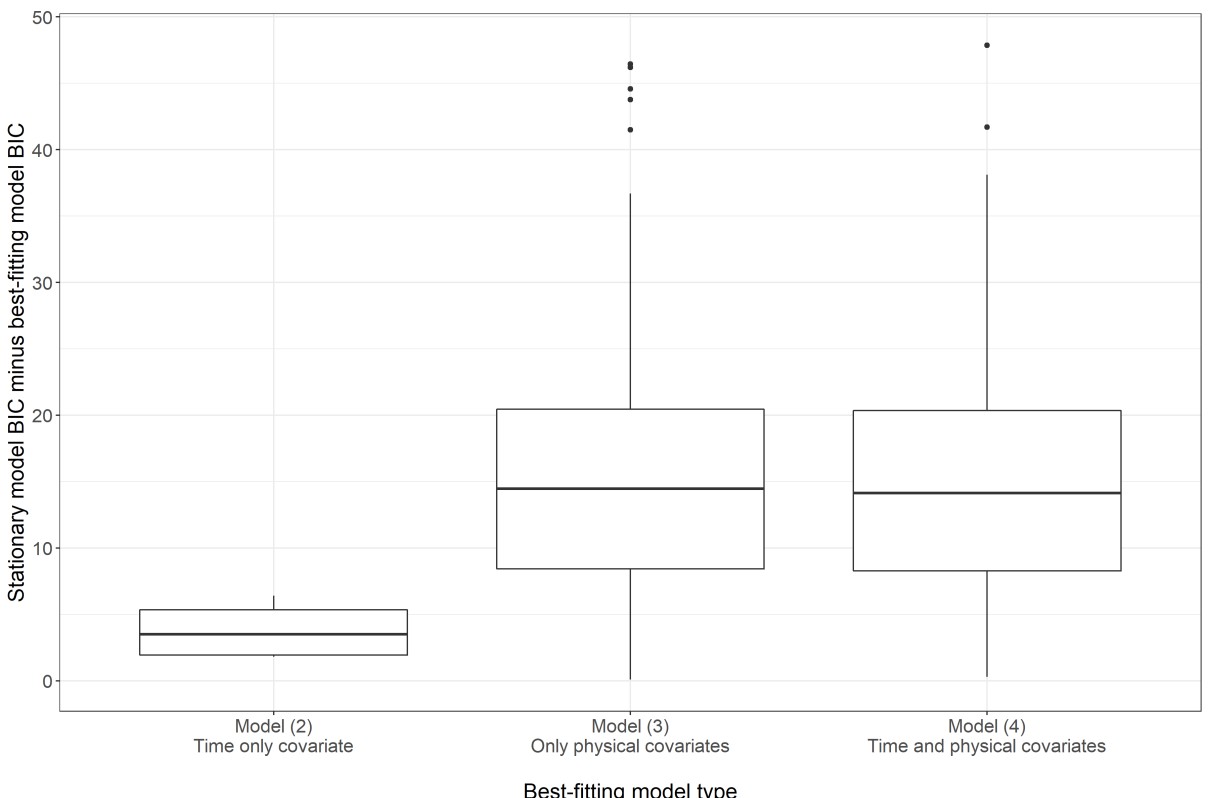

**Figure 1: Boxplots showing the distribution of the BIC improvement between the best-fitting model (where it is non-stationary) and the stationary model**

In cases where a linear trend improves the fit, i.e. where model (4) beats models (1) to (3), this can be interpreted as a sign that other physical factors that affect flood flows are changing over time.

As mentioned above, an additional model version was fitted at sites where version (4) was the preferred fit out of (1) to (4). Contrary to expectations, BIC values for version (5), fitted to a single physical covariate as measured, were not lower than those for version (4) at most sites. It would be expected that, if year-to-year variations were being captured by the detrended physical covariate, the longer-term changes in the measured value of that covariate would also be important. However, it appears that the linear trend in the distribution parameters over time cannot be fully captured by any linear trends embedded in the physical covariates. Further work could look at fitting a sixth model version, in which the covariates are time and a physical covariate as measured. This has potential problems with collinearity as both covariates may contain linear trends, and in Appendix B we suggest a way of overcoming this difficulty.

All the following results are based on models using detrended physical covariates.

We have examined the results for any differences in the physical characteristics of the catchments for gauges that are fitted with model types (3) and (4). A two-sided Student's t-test found no significant differences, at a 5% significance level, in the mean catchment areas, soil types, urban extents, influence of lakes or extent of floodplains. Significant differences, at a 1%





significance level, were evident in the annual average rainfall and mean gradient of the catchments. Gauges fitted with model type (4), where time is a significant covariate, tend to be on steeper catchments, with higher rainfall (Figure 2). This is consistent with findings that temporal trends in peak flows are strong in the north and west of England (Faulkner *et al.*, 2020).

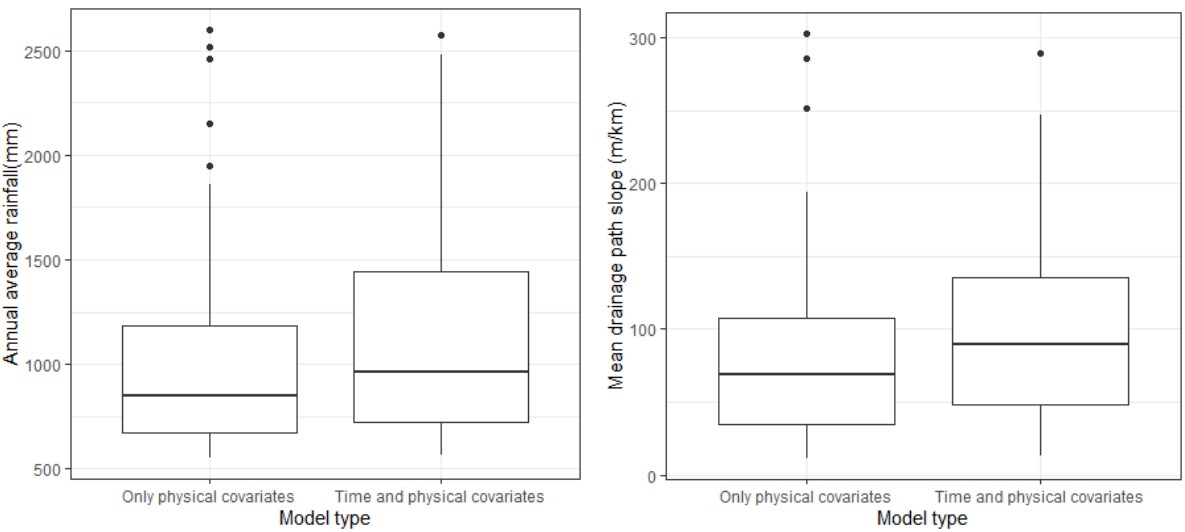

**Figure 2: Boxplots showing the distribution of (a) catchment rainfall and (b) slope for the two non-stationary model types that fit best at nearly all gauges**

When physical covariates are selected, the annual rainfall is by some way the most common choice (Table 2). It is included as a covariate for the location parameter at 65% of all gauges, and for the scale parameter at 21% of all gauges (the scale is 265 modelled as fixed, with no covariate, at most gauges). In some cases, annual rainfall is included alongside time as a covariate, and in others it is the sole covariate.

**Table 2: Most commonly selected covariates (GEV distribution)**

| Covariates for the location parameter | | | Covariates for the scale parameter | | |
|---|---|---|---|---|---|
| Rank | Covariate | % of gauges where chosen | Rank | Covariate | % of gauges where chosen |
| 1 | Annual rain | 65 | 1 | None (i.e. parameter is fixed) | 57 |
| 2 | Time | 28 | 2 | Annual rain | 21 |
| 3 | Winter rain | 22 | 3 | Time | 15 |


| 4 | Autumn rain | 5 | 4 | Winter rain | 4 |

Note: Percentages in the columns can add up to more than 100, because models can include both time
and a physical covariate together.

The second most useful physical covariate was the winter rainfall, chosen at 22% of gauges for the location parameter and 4%
for the scale parameter.

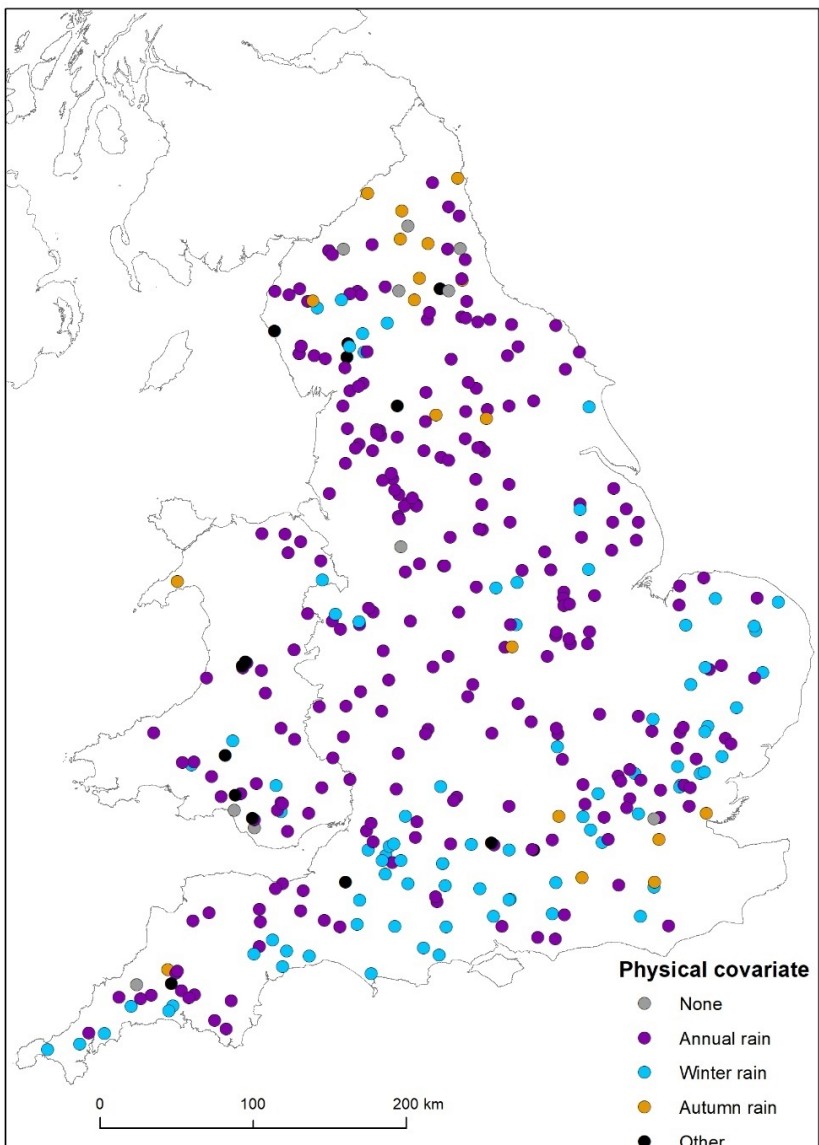

**Figure 3: Best-fitting physical covariate chosen at each gauge. "None" is used when the best-fitting model is stationary.**





Figure 3 shows that, while the annual rainfall is widespread as a preferred physical covariate across the country, the winter rainfall tends to be preferred in some parts of southern England. There appears to be some correspondence between these gauging stations and the locations of chalk outcrops, which are concentrated in central southern England and along a band running from there north-east into East Anglia. This is borne out in Figure 4(a) which shows that catchments with winter rainfall as a covariate tend to have a higher baseflow index, indicating a more groundwater-dominated flow regime. This makes

sense physically: flood flows on groundwater-dominated catchments are expected to be strongly linked with the level of the water table, which is determined mainly by the volume of winter recharge. On chalk catchments, rainfall outside the winter recharge season may be lost due to evaporative demands and so have little impact on flood flows. Chalk aquifers, which have much lower specific yield than sandstone for instance, tend to respond more rapidly to recharge over a single winter season.

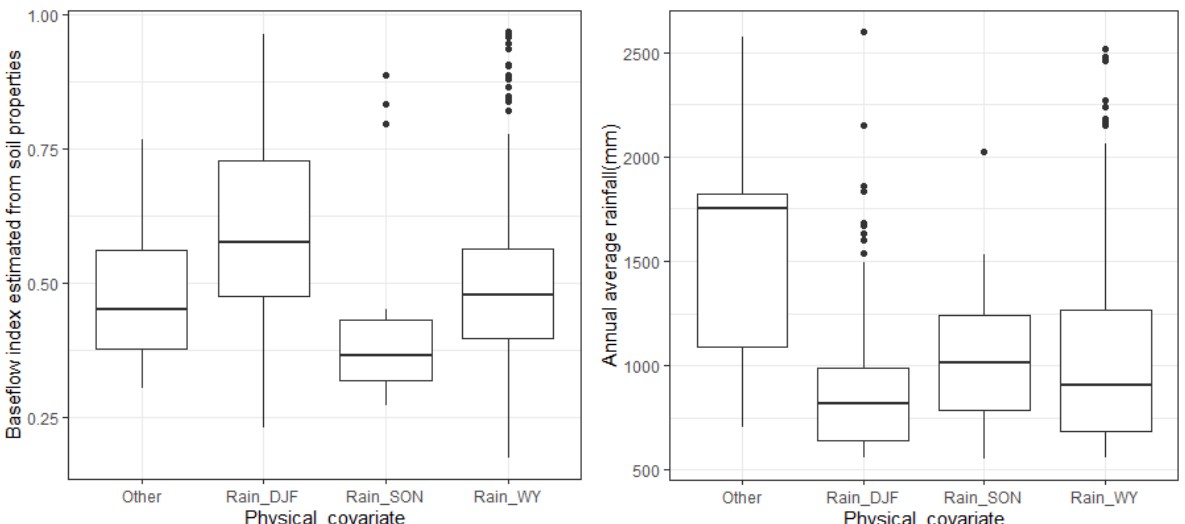

**Figure 4: Boxplots showing the distribution of (a) baseflow index and (b) catchment rainfall for models including different physical covariates. DJF is winter rain, SON is autumn rain and WY is water year rain.**

There is also an association between choice of physical covariate and the average annual rainfall of the catchment (Figure 4(b)). For example, where covariates other than annual, winter or autumn rain are preferred, this tends to be on catchments

with very high rainfall. This is consistent with findings by Chen, Papadikis *et al*. (2021) that annual rainfall tends to be the best covariate in lowland areas of England, with the annual maximum daily rainfall being preferred in the rest of the UK where catchments tend to be dominated by low permeability bedrock and so sensitive to high intensity rainfall.

It is worth noting that the rainfall covariates associated with each annual maximum flow are calculated by accumulating rainfall within the same water year as the annual maximum flow. For annual total rainfall, this means rainfall between 1 October and

30 September. It is likely that some of this rainfall will have occurred after the annual maximum flood. Despite this, the annual rainfall appears to be a widely preferred covariate.



Example results for one gauge are shown in Figure 5. It is common to present results of non-stationary models as a time series, but this is less appropriate for models with physical covariates. Instead the results are shown in the probability domain, as integrated flow estimates for a range of encounter probabilities evaluated over the 48-year length of the gauged record. The

non-stationary flow estimates are distinctly higher than the stationary, especially for low probabilities. They also have much wider confidence limits. The magnitudes of the recorded annual maximum floods are marked on the y axis. The highest flood on record, 125m³/s, is associated with an encounter probability over the record length of 38% according to the stationary results and just under 80% according to the non-stationary results.

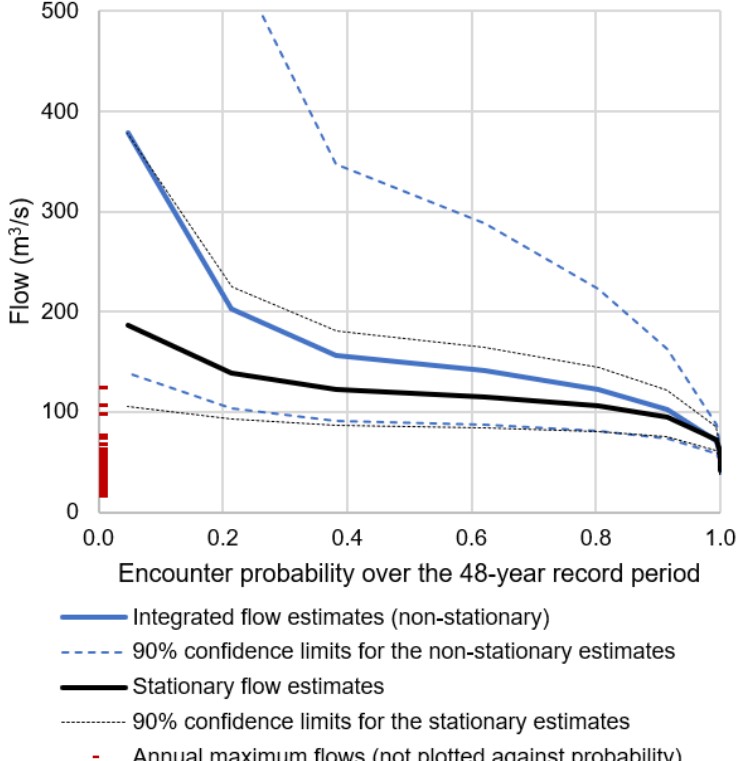

**Figure 5: Example of integrated flow estimates compared with stationary estimates for the Leven at Leven Bridge**


The integrated flow estimate is representative of conditions during the whole period of recorded flow and covariate data, rather than being associated with one particular point in time. The single-year integrated flow estimate can be found when time is included as a covariate and has been estimated for the most recent year in the period of record to provide a near present-day

estimate. Single-year integrated flow estimates have been extracted where available (i.e. where the best-fitting model includes time as a covariate) and integrated flow estimates have been extracted everywhere else for three specified annual exceedance probabilities so that they can be easily compared with stationary estimates (Figure 6). All of these estimates are representative of the end of the record and are therefore comparable.





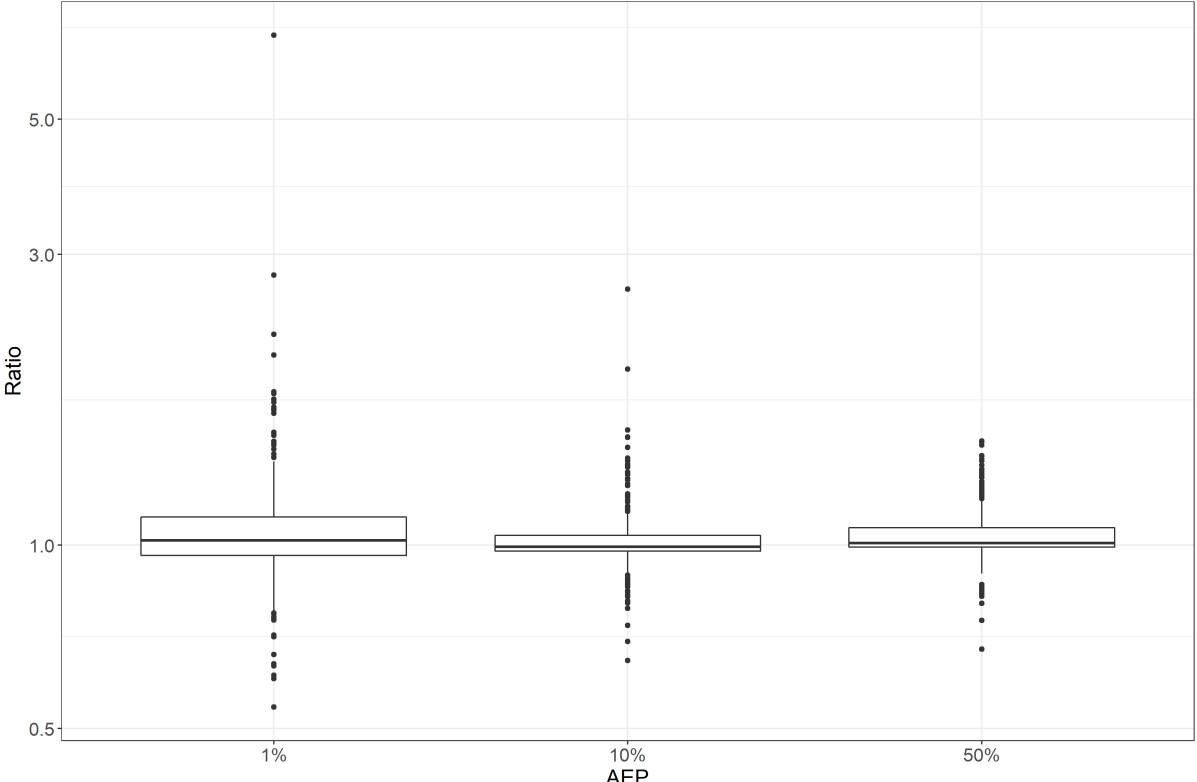

**Figure 6: Box and whisker plot showing ratios of integrated or single-year integrated flow estimates from model with lowest BIC (if non-stationary) to estimate from stationary model. Results are all representative of the most recent year of record.**

For all three AEPs, the median ratio of integrated (or single-year integrated) flow estimate from the best-fitting model (where it is non-stationary) to the flow estimate from the stationary model is close to 1. The ratios can be much larger though, particularly for the 1% AEP, where one non-stationary flow estimate was nearly seven times larger than the stationary estimate. It is also possible that flow estimates from the best-fitting non-stationary models can be smaller than stationary estimates, with the minimum ratio for the 1% AEP being 0.54.

## 5 Discussion and conclusions

This research has made it possible to apply non-stationary flood frequency analysis incorporating physical covariates to practical problems in flood risk management.

We have shown that incorporating physical covariates improves the fit of flood frequency models for nearly all river gauges in England and Wales. At about one third of gauges, the best-fitting model was a non-stationary version that had time as a covariate in addition to detrended physical covariates. By integrating over the distribution of the covariates it is possible to



extract estimates from such models that define the flow for a given encounter probability over the period of record (the
integrated flow estimate). If the integration is only over the physical covariates, the estimate can define the flow for a given
annual exceedance probability in a given year, for example the most recent year of record (the single-year integrated flow
estimate). This can be a useful quantity for flood risk managers or insurers who need an updated estimate of flood hazards that
avoids having to assume stationarity.

For making investment decisions it is also necessary to look to the future evolution of the flood hazard. Schlef *et al.* (2020)
provide a classification of approaches for projecting future flood hazards, including trend-informed, climate-informed and
hydrological simulation. Trend-informed refers to non-stationary statistical models with time as the only covariate. It is
difficult to justify using such models to predict future trends. Both the climate-informed and hydrological simulation
approaches make use of projections from physics-based climate models. The climate-informed approach uses these projections
in non-stationary statistical models with climatic covariates. Hydrological simulation is the conventional approach using
rainfall-runoff modelling with projected climate inputs. Many conflicting claims have been made about the merits of these
approaches, and others such as stationary flood frequency analysis (Schlef *et al.*, 2020).

One argument in favour of the climate-informed approach for estimating future flood frequency is that it can use climatic
variables related to large-scale oceanic-atmospheric patterns, such as mean annual temperature or rainfall, that global and
regional climate models can predict more accurately than local short-duration rainfall intensity (e.g. Sraj *et al.*, 2016, Schlef *et
al.*, 2020). For example, Wasko (2021) suggests that non-stationary models that use climatic covariates offer a way of
estimating future flood frequency, with covariates projected using simulations from global climate models and the covariates
being attributed to observed changes. One such suggested covariate is the product of monthly temperature and monthly rainfall
(Towler *et al.*, 2010).

On the other hand, even with physically plausible covariates, climate-informed statistical models cannot distinguish correlation
from causation (Slater *et al.*, 2021a). Hence, it is necessary to demonstrate a strong causal relationship for physical covariates
if they are to be used for predictions. Claims that models with covariates such as annual rainfall can be applied in conjunction
with projections of future annual rainfall to provide estimates of the future flood frequency curve appear to make an assumption
that future changes in peak flow can be entirely explained by changes in annual rainfall. Although climate change is expected
to affect annual rainfall, and hence catchment wetness, it can also be expected to influence other factors that control flood
magnitudes such as storm tracks, rainfall intensity and evapotranspiration (which influences soil moisture). These effects
cannot all easily be represented by the relatively simple covariates used in the climate-informed approach.

Some similar arguments can be made against the current generation of the hydrological simulation approach. For example, the
effects of climate change on storm intensity are not represented in the hydrological simulations that were used to derive current
guidance on the impacts of climate change on peak rivers flows in England. Kay *et al.* (2021) used changes in monthly rainfall
and potential evaporation, running simulation models at a daily time step and assuming that daily rainfall scales with the
change in monthly rainfall. Since most catchments in England have a critical storm duration of hours up to a few days, changes
in monthly rainfall may not be well representative of the intensification of flood-producing rainfall on most rivers.





The hydrological simulation approach can lead to derivation of change factors which practitioners can apply to estimates of flood frequency made for a baseline period. For example, in England, change factors vary with river basin, with epoch (2020s,
2050s or 2080s) and are provided for a range of percentiles which express some of the uncertainty in the climate projections (Environment Agency, 2021). One difficulty with this approach is that it tends to assume a stationary baseline period, i.e. climate change is treated as a purely future phenomenon. This assumption is difficult to reconcile with the provision of change factors that show (mainly) increases in peak flows for the present epoch, the 2020s, in comparison with an earlier baseline period.

It would be desirable to have a hybrid approach that could draw on the strengths of both the climate-informed and hydrological simulation approaches (Schlef *et al.*, 2020). One possibility would be to use the outputs of hydrological simulation as a covariate in climate-informed statistical models, for example using soil moisture as a covariate (Tramblay *et al.*, 2014). An ideal hybrid approach might seamlessly model both past and projected future non-stationarity, and account for the impact of climate change on localised, short-term rainfall intensity as well as on longer-term rainfall and soil moisture. It should also
allow for the possibility of non-stationarity due to land use change.

There are some advantages to fitting statistical models to the output of physics-based climate models rather than only to observations. These include more confidence in identifying causal relationships and the ability to separate forced and stochastic components of the signal (Vecchi *et al.*, 2011). Slater *et al.* (2021b) promote so-called hybrid statistical-dynamical modelling as a way of taking advantage of the ability of physical models to predict and explain large-scale phenomena and the strengths
of non-stationary statistical models to estimate probabilities of extreme events.

Such a hybrid approach would face some challenges if applied to flood frequency estimation, because it would depend on having a coupled climate-catchment model that was capable of accurately reproducing the evolution of the statistical characteristics of floods over a period of time equivalent to typical gauged river flow record lengths, and into the future as well. To achieve this on many UK catchments the climate model would need to run at a high resolution to resolve local-scale
atmospheric convection. The UKCP CPM (convection-permitting model) (Chen, Paschalis *et al.*, 2021) meets this requirement and is capable of running for time slices covering several decades. The statistical component of a hybrid model might benefit from including a covariate that functions as an indicator variable to distinguish between observed (past) and modelled (future) data. Brown *et al.* (2014) applied this technique when developing a non-stationary statistical model fitted to both observed rainfall and modelled projections of future rainfall. The indicator variable introduced an opportunity for bias correction, after
which global temperature was introduced as a covariate.

The methods described in this paper are now available to practitioners in the form of a user-friendly R package and guidance document, and are being used in the planning and design of flood alleviation schemes in England.



## Appendix A: Confidence limits for non-stationary flood frequency models

Confidence limits can be calculated using the parametric bootstrapping procedure, similar to that proposed by Eastoe and Tawn
(2009). The parametric bootstrap is a general algorithm for deriving the sampling properties of estimators of parameters using a selected inference method. It is a three-step process: (i) generating data from the fitted statistical model to reflect the sample size and variations of the observed data, (ii) using the same inference method on the simulated data whilst assuming that the parameters used in this simulation are unknown, and (iii) repeating the first two steps many times.

To help understand the algorithm first consider an independent and identically distributed set of data with each value believed
to come from a parametric distribution function $F(y; \theta)$, described by unknown parameters $\theta$. In what follows we denote by $\hat{\theta}$ an estimate of the parameters derived from the observed sample $y_1, \dots, y_n$.

1. Generate a data sample of size $n$ from the fitted model, i.e. using $F(y; \hat{\theta})$.

2. Fit the model for the data simulated in step 1 to give a new estimate $\hat{\theta}^{(1)}$.

3. Repeat steps 1 to 2 $k$ times to give a set of estimates $\hat{\theta}^{(1)}, \dots, \hat{\theta}^{(k)}$, called a bootstrapped sample.

4. Use the sample of $k$ estimates to derive the confidence interval for each element of $\theta$, by ranking the bootstrapped sample for each element and picking, for example, the 2.5% and 97.5% quantiles to give a 95% confidence interval.

5. If the interest is in some function of $\theta$, say $g(\theta)$, then the bootstrap sample $g(\hat{\theta}^{(1)}), \dots, g(\hat{\theta}^{(k)})$ can be used to construct the confidence interval for $g(\theta)$.

When there are covariates in the model, such as the detrended physical covariates or water year, here denoted by $x$, the
data $y_1, \dots, y_n$ are no longer identically distributed over observations. Despite this the parametric bootstrap is still straightforward to apply as long as the $y_1, \dots, y_n$ are independent given the associated covariates $x_1, \dots, x_n$.

For conditional flow estimates, covariates remain fixed in step 1 rather than being resampled prior to estimation of the model parameters $\hat{\theta}^{(i)}$. The sample to build confidence intervals for the conditional estimates is $\hat{y}_p^{(i)}(x) = F^{-1}(p|x; \hat{\theta}^{(i)})$ for $i = 1, \dots, k$.

For integrated flow estimates, the covariates are sampled first (with replacement or from $\hat{f}(x)$) and then the data are simulated conditional on the covariates. There are two sources of uncertainty in the integrated flow estimates: the parameter estimates $\hat{\theta}$ and the estimated distribution of the covariates $\hat{f}(x)$. Firstly, ignoring the latter source gives $\hat{y}_p^{(i)} = F^{-1}(p; \hat{\theta}^{(i)})$ for $i = 1, \dots, k$, from which confidence limits are derived as in step 5 above.

If the uncertainty in the distribution of the covariates is to be included, we need to use the sample of distribution estimates
$\hat{f}^{(1)}, \dots, \hat{f}^{(k)}$, either from the same bootstrap sample or from some independent source if the covariates are being set to represent a future period.

The following is then obtained:

$$F(y; \hat{\theta}^{(i)}, \hat{f}^{(i)}) = \int_{\Phi} F(y|x; \hat{\theta}^{(i)}) \hat{f}^{(i)}(x) \, dx \quad \text{for } i = 1, \dots, k$$





where in each of the $k$ bootstrap samples, uncertainty in the covariate distribution is incorporated in the uncertainty of the
estimated model parameters via the sampled covariates in step 1, and via inclusion of the same sampled covariates in the terms
in the above integral. The resulting sample of estimates from the above equation is used to build confidence limits in the usual
way.

$$\hat{y}_p^{(i)} = F^{-1}(p; \hat{\theta}^{(i)}, \hat{f}^{(i)}) \text{ for } i = 1, \dots, k$$

The uncertainty in $\hat{f}(x)$ can be ignored when time is the only covariate, since $f(x)$ in that case forms a uniform distribution
with no uncertainty. One challenge with this approach is that it involves many calls to the MLE function, some of which may
not converge.

**Appendix B: A method of avoiding problems with collinearity when using both time and physical covariates**

For convenience we re-state here the various types of models discussed in the paper:

(1) a stationary version,
(2) versions with just time as a covariate,
(3) versions with just one detrended physical covariate, and
(4) versions with both time and a detrended physical variable.

To better understand the interaction between time and physical covariates and pave the way to a more causal explanation of
non-stationarity, it is desirable to fit two additional model versions in which the physical covariates are not detrended:
(5) versions with just one physical covariate, as measured
(6) versions with both time and a physical covariate as measured.

Model (6) has potential problems with collinearity as both covariates may contain linear trends. To tease out the relative
importance of the two covariates it is important to overcome the collinearity. We present here a potential solution.

Suppose that model (4) fits best out of models (1) to (4). Say that one of the parameters of model (4) (location or scale) is
given by:

$$\alpha_0 + \alpha_1 t + \alpha_2 z_t$$

where $t$ is time (year), $z_t$ is the detrended physical covariate at time $t$, $\alpha_0, \alpha_1, \alpha_2$ are coefficients.

Due to the detrending this parameter can also be expressed as:

$$\alpha_0 + \alpha_1 t + \alpha_2 (x_t - \gamma t)$$

where $x_t$ is the as-measured physical covariate at time $t$, $\gamma$ is the gradient in the detrending of $x_t$.

By rearrangement this is equal to

$$\alpha_0 + (\alpha_1 - \gamma \alpha_2) t + \alpha_2 x_t \tag{B1}$$

so is a linear relationship in terms of time and the measured physical covariate.



If the term $(\alpha_1 - \gamma\alpha_2)$ were equal to zero then the only covariate would be $x_t$, the measured physical covariate. This is model (5). However, our results indicated that model (5) did not fit as well as model (4) at most sites, which suggests that $(\alpha_1 - \gamma\alpha_2)$ differs from zero.

The above mathematics gives a helpful insight for model (6), for which the same parameter can be expressed as a function of time and the measured physical covariate:

$$\delta_0 + \delta_1 t + \delta_2 x_t$$

where the coefficients are now $\delta_0, \delta_1, \delta_2$.

If fitting model (6), then one should fix $\delta_2$ equal to the estimate we get for $\alpha_2$ from model (4), where there were no issues about collinearity in the two covariates. We can see in Equation B1 that without collinearity $\delta_2$ must be this value. This means that only $\delta_0$ and $\delta_1$ need to be estimated when fitting model (6). Then $\delta_1$ gives the residual linear trend component of the non-stationarity not accounted for in the physical covariate $x_t$.

If the resulting estimate of $\delta_1$ is nearer to zero than was $\alpha_1$ when estimated in model (4), this would indicate that some of the long-term linear trend in flood magnitudes has been explained by long-term changes in the physical covariate, while avoiding any difficulties of collinearity.

This additional fitting step is recommended for any further investigations.

**Code availability**

The procedures for non-stationary flood frequency estimation are implemented in the R package *nonstat*, available from https://www.gov.uk/flood-and-coastal-erosion-risk-management-research-reports/development-of-interim-national-guidance-on-non-stationary-fluvial-flood-frequency-estimation.

**Data availability**

The dataset of annual maximum flows for England and Wales was based on the National River Flow Archive dataset available
from https://nrfa.ceh.ac.uk/peak-flow-dataset. The version of the dataset used for the analysis is available from the authors. The results of the national analysis are available from https://www.gov.uk/flood-and-coastal-erosion-risk-management-research-reports/development-of-interim-national-guidance-on-non-stationary-fluvial-flood-frequency-estimation.

**Author contribution**

SL commissioned and reviewed the research, DF led the research team, SW developed the code and JT advised on statistical
aspects. DF prepared the manuscript with contributions from all co-authors.

The authors declare that they have no conflict of interest.





**Acknowledgements**

The development of the methods described here was funded by the Environment Agency. Peak flow data were obtained from the UK National River Flow Archive and the Environment Agency.

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
