# Peer review of "Modelling non-stationary flood frequency in England and Wales using physical covariates"

_Hydrology and Earth System Sciences, 2022_

## Author Comment (AC1)

https://hess.copernicus.org/preprints/hess-2022-401/

Review comment posted 8 Feb 2023, with responses from the authors.

Thank you for the opportunity to review Faulkner et al. Modelling non-stationary flood frequency in England and Wales using physical covariates

We are grateful for this in-depth and carefully considered review.

The central problem the manuscript seeks to address (that of extracting decision-useful outputs from non-stationary flood frequency analysis) is compelling problem and I agree with the authors that this issue is often overlooked in the academic literature. The fact that one of the authors is from the Environment Agency of the UK shows that this work is recognized by the appropriate government agency.

That being said, this manuscript is essentially a rewrite of a technical report on the topic of non-stationary flood frequency submitted to the Environment Agency (Faulkner et al., 2020; which is extensively referenced in the manuscript). I did not do an extensive/exhaustive comparison, yet it is clear that some portions of the manuscript are directly copy-paste (or nearly so; e.g., compare lines 81-105 of the manuscript to pgs. 34-35 of the report, lines 119-125 of the manuscript to pgs. 36-37 of the report, Figure 3 of the manuscript to Figure G-9 of the report, and Table 2 of the manuscript to Table 4.7 of the report). I state this (without any judgement) for the purpose of bringing it to the attention of the editor.

It is correct that parts of our manuscript are closely based on the report that we wrote for the Environment Agency, as we state on lines 53-55. The report covered several aspects of non-stationary flood frequency estimation, and in this paper we bring out one, that we believe represents a significant advance and one that is worth wider dissemination. In the paper we add, among others, further development of the concept (e.g. version 5 of the candidate models), a more focused and up-to-date literature review and a wider-ranging discussion of how this sort of analysis can be extended to consider future conditions.

Apart from what is noted in the previous paragraph, the science presented in this paper is incremental. Essentially, this is a case study applying non-stationary frequency analysis to flood events in England and Wales.

The primary advancement, that of an "integrated flow estimate" which allows for representing the non-stationary results in a decision-centric way, is essentially an integration of the non-stationary flood frequency estimate over the domain space of the covariates. I think this advancement, while incremental, is useful and should be made know to the academic/practitioner world. However, it seems it has already been proposed by Eastoe & Tawn (2009). If improvements are needed (lines 127-129), then perhaps this additional paper is justified; but, what were the weaknesses? (They are not mentioned) and does this paper address those weaknesses? Here was a missed opportunity to further justify the significance of this paper.

We acknowledge that on a probabilistic level the advance is incremental. However, from the perspective of statistical application, this is the first application of the concept of the marginal return level to flood frequency analysis. The covariates used by Eastoe and Tawn (2009) primarily exhibited seasonal variation and longer-term trend was not considered. This was one reason given by Faulkner et al. (2020) as to why further investigation is needed before physically meaningful covariates can be applied in flood management practice. Another was the need to consider how to incorporate covariates that have distributions with tails that are not well-represented by empirical estimates, e.g., can have values much larger than the maximum observed values.

An additional innovation is the idea of integrating over some but not all covariates, so that the estimate from the model remains conditional on some covariates, such as the year (the "single-year integrated flow estimate"). We will expand the discussion of this because it can be generalised to any covariate whose behaviour is reasonably predictable such as urban cover.

However, the reason that most convinces us that this advancement should be published in a hydrological journal is our impression, discussed in our introduction, that papers on non-stationary flood frequency typically do not include the step that would allow results to be extracted from non-stationary models with physical covariates.

We will add more justification of the significance of the paper in accordance with these suggestions.

Also, the results associated with this "integrated flow estimate" are fairly minimal (Figures 5 and 6; lines 297 – 322; nothing at all in the discussion and conclusion). Since this is the stated main point of the manuscript, more is needed.

You could walk through a hypothetical example of how a decision-maker or engineer might use Figure 5. You could answer questions such as (and I am sure you can think of additional ones to enrich this portion of the manuscript): What is the implication of a 7x ratio of non-stationary to stationary? Which AEPs are used by UK regulatory standards (and how that does affect the interpretation of these results)? What about the case where the ratio was 0.54 (should we disregard the non-stationary model in that case and use the stationary model since it is more conservative)?

The main point of the paper is to present and discuss the method rather than its results from one particular application, which is included as an illustration, as we explain in lines 51-53. However, we are happy to add more discussion of how the results might be interpreted and applied. One aspect we can emphasise is that there are other factors beyond the model fit as measured by BIC that should be considered when selecting a preferred model for decision-making at a particular site. These include other statistical measures such as AIC and likelihood ratios, inspection of model fit on P-P and Q-Q plots, consistency of model form between locations and hydrological reasoning, confidence limits and comparison between the flow estimates and the recorded flood peak data. We would recommend that practitioners consider all these in combination.

Which brings me to the issue that the discussion and conclusion seem disconnected from rest of paper. They read more like a literature review which is found at the beginning of a work, rather than a reflection on what was done. I do not disagree with what is noted in the discussion and conclusion, but it is not in the right place in the manuscript and is not connected with what was shown in the results. Consequently, the reader misses out on a discussion on the actual results that were presented (e.g., answers to the various questions posed in the previous paragraph).

The majority of the discussion section (lines 334 to 390) is about extending non-stationary models into the future. The reason why we included this material in the discussion rather than the introduction is that it is pointing ahead to potential future work, supported by discussion of the current state of scientific practice, rather than introducing the background to the analysis presented in the paper. However, we acknowledge it is unconventional to include so much literature citation in the discussion so we will look to find a more appropriate home for this material in the paper.

Some minor comments

Line 79: To my knowledge, I do not think that Francois et al. 2019 use/reference the East Atlantic pattern (East Atlantic is not found when I do a search). Please double check and correct the reference. As a recent review paper on this topic, Francois et al. 2019 is relevant and should probably be referenced in this manuscript, but referenced correctly.

Correct, the reference about the East Atlantic pattern should have been to Steirou et al. (2019), as for the NAO. We agree Francois et al. is an important paper and will refer to it.

Line 82: AMAX is not defined.

We will spell this out.

Lines 94 – 105: I am not sure that this discussion of Reason 2 for choosing physically-based covariates correctly captures the intent of many of the authors I have read, and the approach as best implemented. The intent of a physically-based covariate to is represent mathematically some physical driving mechanism of floods, whether oceanic, atmospheric, or land based. The confusion of correlation for causation will only happen when an analyst somewhat blindly applies this method: that is, coming up with a suite of possible covariates and trying as many possible and choosing the one with the best correlation. As written, I think this section is somewhat misleading (as if it is the fault of the method, when really this is an error in application of the method).

We have encountered some confusion between correlation and causation in situations other than that suggested. Even covariates that represent a plausible physical driving mechanism may not necessarily be the sole or main driving mechanism for a particular catchment, but if the covariate is increasing along with the floods, there may be a correlation. We will add reference to the three ingredients of trend attribution

suggested by Merz et al. (2012) : evidence of consistency, evidence of inconsistency, and provision of confidence level. The same paper makes some specific points about correlation between precipitation and flood magnitude being an insufficient way of identifying a driving mechanism.

We could also cite the work of Montanari and Koutsoyiannis (2014) and Serinaldi, Kilsby, and Lombardo (2018) who argue that a non-stationary model can only be justified where one has deterministic information on the process of change.

We agree that such confusion is not the fault of the method, and will make this clearer. In our experience it is a very common misconception in the method's application. The point that we make that "in principle it would be possible to include any covariate with a trend, whether or not it had any physical connection with the processes that cause floods." is deliberately absurd and so perhaps too easily dismissed. We will strengthen the argument by discussing a more realistic example, which is already outlined in lines 349-356 in the discussion section.

Line 119: The flood frequency estimate is time-dependent regardless of the covariates – either directly dependent on time or implicitly dependent on time via a physical covariate. Wording could be improved here.

Agreed, we will make this change.

Line 198: Why were the models just limited to two covariates? In particular, why weren't the two physically based covariates used in one model – are they too strongly correlated? Need to justify this choice. Based on skimming the report, I suspect your answer might be that it was too many models... perhaps; but the model with the two physical covariates seems important enough to test regardless of the issue of computation.

Both suggested reasons are relevant: cross-correlation of the physical covariates and proliferation of model types. We regarded 88 candidate models as a large enough number to consider. For other applications in which covariates describing catchment land cover are combined with climatic covariates, it may be reasonable to allow combinations of physical covariates. We will add this point to the paper.

Line 219: Why include the GLO if the results are not analyzed? My suggestion is to remove it from the paper, or justify its inclusion (and include results).

We are happy to remove reference to the GLO.

Figure 6: They y-axis scale is confusing. Is it a log-scale? If so please note in the caption. More labels would also be helpful.

We will improve this figure.

Appendix B: Why is this included? It is just a suggestion, without any testing or analysis or use in the manuscript. I suggest to remove it or to more fully incorporated in the manuscript.

We included it in the hope that others could use in future work. But we are willing to remove it.

**References**

Merz, B, Vorogushyn, S, Uhlemann-Elmer, S., Delgado, J and Hundecha, Ya. (2012) More Efforts and Scientific Rigour Are Needed to Attribute Trends in Flood Time Series. *Hydrol. and Earth System Sci.,* 16, 1379-1387.

Montanari, A. and Koutsoyiannis, D. (2014) Modeling and mitigating natural hazards: Stationarity is immortal!, Water Resour. Res., 50, 9748–9756.

Serinaldi, F., Kilsby, C.G. and Lombardo, F. (2018) Untenable nonstationarity: An assessment of the fitness for purpose of trend tests in hydrology. Advances in Water Resour., 111, 132-155.

---

## Author Comment (AC2)

https://hess.copernicus.org/preprints/hess-2022-401/

Review comment posted 20 Feb 2023 by Kolbjorn Engeland, with responses from the authors.

The paper addresses the challenge of non-stationary flood frequency modelling and how to make such models useful for decision makers. In general, the paper is well written and could be published (provided it is different enough from Faulkner et al., 2020).

Thank you very much for taking the time to comment on our paper. We are very grateful for your insights.

The paper is rather short, and have a lot of questions after reading the paper. I therefore think there is room for several types of improvements. Below are some suggestions.

1: I think the numbering of sections might be improved. In particular, the heading on line 55 does not belong to the introduction section since the following paragraphs describe the methods applied in this paper. One solution is to create one section 'Methods' that contains lines 56 – 176.

Thank you for pointing this out.  We will improve the numbering.

2: No results are shown for the GLO distribution, so it is not necessary to include it at all in the paper.

Agreed, and the first reviewer agrees too. We will remove it.

3: I am not completely convinced of the usefulness of Model 3 where de-trended physical covariates are included.

We agree that the justification for Model 3 needs to be strengthened.  We see it more of a stepping stone, an aid to understanding, rather than an end in itself. We propose to add some text to make clearer what the value of Model 3 is, namely that it can help to disentangle the effect of any long-term trend from shorter-term cycles or fluctuations in the physical covariates. Comparing the various models helps us to assess if it is the specifics of the physical covariate or simply that it is may be approximately linear over time which is causing the physical covariate to appear statistically significant.

Model 3 turns out to be the best-fitting out of 1-4 at many gauges, and we could hypothesise (and will test this) that these tend to be gauges without significant long-term trends in peak flows. There may still be cyclical non-stationarity at some of these gauges that is being described by cycles in the covariates. However, as we have written, it would seem odd if year-to-year variations were captured by a detrended physical covariate without the longer-term changes in that covariate also being important, so we will add Model 5 into the comparison of Models 1-4.

Another possibility at some sites is that there is little trend present in the covariate and so Model 3 is similar to Model 5.  As you have suggested, we will examine the extent of trends in the covariates.

We address some of your follow-up parts to this comment briefly below.

Firstly, I cannot see how the covariates are detrended: for which period was the trend calculated, and did you use a linear trend?

We will add to the paper: it was a linear trend over the whole period of record.

Secondly, how many of the physical covariates had a significant (and substantial trend) where model 3 and 5 were different ?

This is something that we could check and add to the paper.

Thirdly, using a de-trended covariate indicate that there is an interaction between time and the physical covariate or potentially interactions with other physical covariate that you have not included.

Agreed (we have said something similar in the paper) – and this seems a reason to avoid preferring model (3), even if it does show some good fits.

Finally, using a detrended covariate will make it difficult to apply the model for a climate in the coming decades.

This is correct. However we do not necessarily recommend using any of these models to estimate future flood frequency, for reasons explained in the discussion section.

This choice of using de-trended covariate need a better explanation and discussion. I think it might be helpful to add Model 6 and the methods described in Appendix B. An alternative solution to using independent covariates is to use regularization methods similar to Lasso regression.

We would have liked to add Model 6 but reluctantly decided we would need to leave that for future work, having already spent far longer than we planned in writing this paper!

Thank you for the Lasso suggestion, which could be a promising way of fitting more complex models without having to try large numbers of combinations of covariates. We will mention regularisation methods in the discussion. We anticipate that Lasso could still have problems picking between casual and non-casual covariates when they are all strongly co-linear.

4: You included trends in both location and scale parameters of the GEV distribution. Did you systematically evaluate all combination of trends in scale and location parameters?

Yes, we included all combinations of trends in both parameters.

Could different physical covariates be selected for the location and scale parameters?

No, we only included one physical covariate at a time. We do not think it would be sensible to have different covariates in the scale that are not in the location, given the theoretical way the these two parameters link to the properties of the core underlying distribution of the hourly river flow data. We will make this clear.

Are the results in Figure 2, Figure 3 and Figure 4 based on the best fitting covariate for the location or scale parameters? Some more details are needed.

The goodness of fit is judged using the BIC which measures the model as a whole rather than the individual parameters.

5: What are the signs of the detected trends (or regression coefficients)?

We agree this is a question that could interest readers and will add this information.

6: I think more results similar to those shown in Figure 5 could be produced. It could be good to show one more plot where floods for the non-stationary model is smaller than the floods for the stationary model. It could also be interesting to see this plot for a model where time is included as a covariate and a model where time is excluded as a covariate.

We can add and discuss some more example plots.

7: Could it be useful to see the results in Figure 6 on a map in order to highlight where the ratio is smaller and larger than 1? Have all records the same year as the last year in the records? If not, how much might the results be influenced by the end year of the record?

This type of information is given in Faulkner et al. (2020) and we will point readers to it.

8: Is it possible to detect more results from the data used to produce Figure 6? In particular, I would like to know for which models or catchments the ratios are far from one. Is it in catchments where the selected models include time as a covariate or are the specific catchment properties or geographical locations that might explain the differences in ratios? Another possibility is that the estimate of the shape parameter are different in the stationary and non-stationary models.

We can add some discussion of this to the paper.

The maximum likelihood estimator is know to results in shape parameter estimates that are not robust, in particular for short record lengths. Could the results you get depend on record length? A penalized maximum likelihood estimator is often recommended for the GEV distribution. Alternatively, a Bayesian approach with a prior on the shape parameter could have been used. I think the robustness of the ML estimator should be discussed.

This is a good point. We did try a penalised ML estimator and found that this approach, at least as implemented in the extRemes package, was unexpectedly counterproductive, leading to some large increases in the shape parameter. We will comment on this and possible future steps to improvement.

9: Uncertainties in the estimates are not discussed. I think this is important since the inflation in uncertainty is a drawback of non-stationary modelling. Figure 5 shows that he uncertainty in the non-stationary model is high, and even if the difference in integrated flow estimate is substantial, the difference is not necessarily significant. Figure 6 shows that a large par of the ratios are close to 1. Is it possible to use the results from all 375 flow records to summarize in how many cases the non-stationary integrated flow estimates are outside the confidence intervals for the stationary model?

We agree that this inflation in uncertainty is important to acknowledge. Thank you for the suggestion for analysis of confidence intervals, which we will look into.

10: The implications for practical applications could be discussed more. Would you recommend to always use the non-stationary model? What are the recommendations if the non-stationary model results ins smaller design floods or in substantially higher design floods (up to 5 times higher than for a stationary model.) ?

We will add some discussion of this, which was also requested by the first reviewer.